# Transposition-mediated DNA re-replication in maize

**Jianbo Zhang[1,2][†], Tao Zuo[1,2][†], Dafang Wang[1,2], Thomas Peterson[1,2]***

[1]Department of Genetics, Development and Cell Biology, Iowa State University, Ames, United States; [2]Department of Agronomy, Iowa State University, Ames, United States

**Abstract** Every DNA segment in a eukaryotic genome normally replicates once and only once per cell cycle to maintain genome stability. We show here that this restriction can be bypassed through alternative transposition, a transposition reaction that utilizes the termini of two separate, nearby transposable elements (TEs). Our results suggest that alternative transposition during S phase can induce re-replication of the TEs and their flanking sequences. The DNA re-replication can spontaneously abort to generate double-strand breaks, which can be repaired to generate Composite Insertions composed of transposon termini flanking segmental duplications of various lengths. These results show how alternative transposition coupled with DNA replication and repair can significantly alter genome structure and may have contributed to rapid genome evolution in maize and possibly other eukaryotes.

## Introduction

Initiation of DNA replication in eukaryotic cells is controlled by the replication licensing system (*Blow, 1993*; *Blow and Dutta, 2005*; *Truong and Wu, 2011*), which ensures that each segment of the genome is replicated only once per cell cycle. The expression and activity of the replication licensing factors are precisely regulated, and misexpression or mutation of these factors can lead to DNA re-replication, genome instability, major chromosomal rearrangements, and tumorigenesis (*Melixetian et al., 2004*; *Green and Li, 2005*; *Rice et al., 2005*; *Hook et al., 2007*; *Liontos et al., 2007*; *Sugimoto et al., 2009*; *Green et al., 2010*). Misregulation of some histone methyltransferases can also result in DNA re-replication in plants and animals (*Jacob et al., 2010*; *Tardat et al., 2010*; *Fu et al., 2013*).

Although DNA replication is strictly controlled, some DNA segments can escape this restriction and replicate more than once in a single cell cycle in normal cells. For example, some Class II DNA transposons, including the maize *Ac/Ds* system, *E. coli* TN*10*, and *E. coli* TN*7*, are known to transpose during DNA replication (*Roberts et al., 1985*; *Chen et al., 1987*; *Peters and Craig, 2001*). If a replicated transposon excises and reinserts into an unreplicated site, the transposon can undergo one additional replication in the same S phase; the re-replication, however, is limited to the TE itself and does not extend into the TE-flanking regions.

We and others have previously shown that a pair of *Ac* termini in reversed orientation can undergo transposition, generating major chromosomal rearrangements such as deletions, inversions, permutations, duplications, and reciprocal translocations (*Zhang and Peterson, 2004*; *Zhang et al., 2006*; *Huang and Dooner, 2008*; *Zhang et al., 2009*, *2013*); this transposition reaction is termed reversed *Ac* ends transposition (RET). All the RET-generated genome rearrangements described to date are fully explained by models in which the excised TE termini inserted into target sites that had completed DNA replication. However, it seems reasonable to expect that RET, like standard *Ac/Ds* transposition, may also occur during DNA replication, and that the excised reversed *Ac* termini could insert into unreplicated target sites. Here, we show that such events do occur, and that they can induce re-replication of the TE and its flanking sequences. This process generates novel structures termed Composite Insertions (CIs) that contain TE sequences and variable lengths of the flanking genomic DNA.

***For correspondence:**
thomasp@iastate.edu

[†]These authors contributed equally to this work

**Competing interests:** The authors declare that no competing interests exist.

**Reviewing editor**: Bin Han, National Center for Gene Research, China

**eLife digest** To make accurate copies of its genome, a cell takes precautions to make sure each section of DNA is only duplicated once in every round of copying. However, there are some sections of DNA called transposons that can avoid these restrictions and be duplicated more often.

Transposons are mobile pieces of DNA: they can be 'cut' from one section of the genome and are able to 'paste' back in somewhere else. The amount of mobile DNA in a genome varies a great deal between species, and in the crop plant maize, it makes up nearly 85% of the genome.

Some transposons can move while the genome is being duplicated. If a transposon is cut out of a section of DNA that has already been copied and is pasted into a site that is yet to be copied, the transposon can be copied again. The transposon may now be present in two different places in the genome.

If two transposons are close together on a section of DNA, both transposons can move at the same time. As they move, they can carry along pieces of the genome, transferring them from one site to another. These transferred pieces can include sections of, or even entire, genes. This is called alternative transposition, but it is not clear whether this process can happen when the genome is actively being duplicated.

Here, Zhang et al. studied transposons in maize. The experiments found that alternative transposition can take place between a site that has already been copied and another site that is still waiting to be copied. Therefore, after the round of copying is completed, both transposons and the flanking DNA can be present in two places in the genome.

When single transposons move, or alternative transposition takes place, sections of the genome can be rearranged and genes can be deleted or new ones can be created. Therefore, transposons may have contributed to rapid evolution in maize and possibly other species.

## Results

### Model of transposition-mediated DNA re-replication

The allele *P1-ovov454* (GenBank accession # KM013692) carries an intact *Ac* element and a fractured *Ac* (*fAc*) element inserted in the second intron of the maize *p1* gene; the 5' terminus of *Ac* and the 3' terminus of *fAc* are present in reversed orientation with respect to each other and separated by an 822-bp inter-transposon segment (*Figure 1A*) (*Yu et al., 2011*). Our recent work showed that the *P1-ovov454* allele undergoes RET to generate derivative alleles containing either deletions or Tandem Direct Duplications (TDDs; *Figure 1—figure supplement 1*). These are formed as a direct consequence of transposition of the *Ac/fAc* termini into a replicated target site on the sister chromatid (*Zhang et al., 2013*). The deletions and TDDs vary in size depending on the position of the insertion site (green/black triangle in *Figure 1—figure supplement 1*); the TDDs previously characterized range in size from 8 kb to 5.3 Mb (*Zhang et al., 2013*).

Here, we asked: what are the consequences of RET events that occur during DNA replication? We developed and tested models in which replicated *Ac/fAc* termini are excised by RET and inserted into unreplicated target sites. As shown in *Figure 1* (See also the animation *Video 1*), this type of transposition reaction places already-replicated DNA in front of a replication fork where it may undergo a second round of replication. We propose that the re-replication fork may spontaneously abort, yielding two chromatid fragments terminated by double-strand breaks (DSBs); fusion of the DSBs restores the chromosome linearity and generates CIs containing *Ac/fAc* and their flanking sequences at the duplication breakpoints. By comparing RET events involving insertion sites that are unreplicated (*Figure 1*) vs replicated (*Figure 1—figure supplement 1*), we can see that both types of events generate TDDs whose sizes are determined by the transposon insertion site. However, only events with unreplicated insertion sites also generate CIs via re-replication of the *Ac/fAc* and their flanking sequences; the resulting products are termed TDDCI alleles. Because the formation of TDDs was described in detail previously (*Zhang et al., 2013*), here we will focus on the origin and characterization of the CI of the TDDCI alleles.

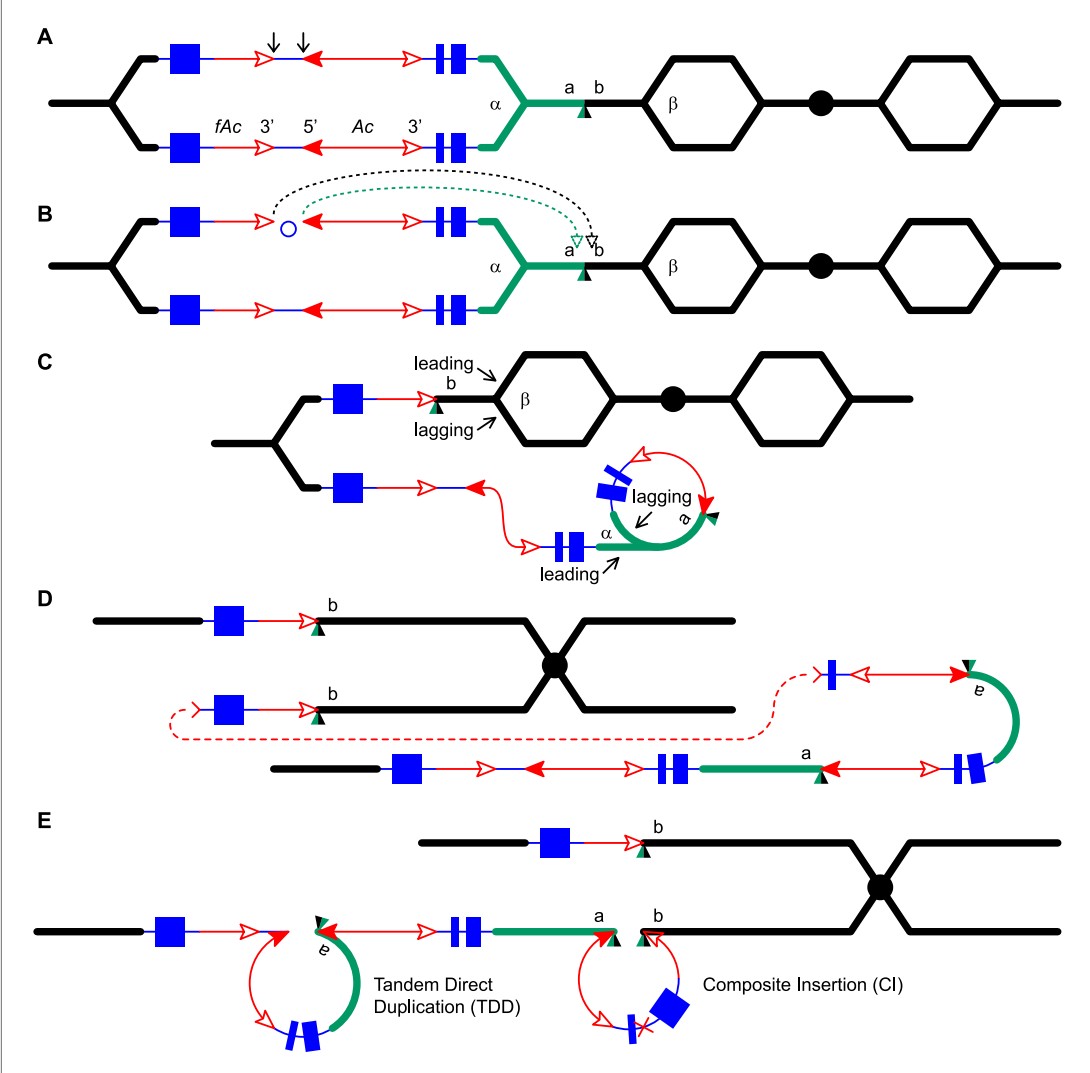

**Figure 1**. Reversed *Ac* ends transposition (RET) during DNA replication generates Tandem Direct Duplication (TDD) and Composite Insertion (CI). Lines indicate a replicating chromosome, hexagons indicate replicons. The blue boxes are exons 1, 2, and 3 (right to left) of the *p1* gene, and the green/black triangles are the transposition target site. Red lines with arrow(s) indicate *Ac/fAc* insertions, and the open and solid arrowheads indicate *Ac/fAc* 3′ and 5′ ends, respectively. Two replication forks considered here are marked α and β. For animated version, see **Video 1**. (**A**) The locus containing *fAc/Ac* is replicated. Vertical arrows indicate the sites of *Ac* transposase cuts at the *fAc* 3′ and *Ac* 5′ ends. (**B**) Transposase cleaves and the inter-transposon segment is ligated to form a circle. The excised transposon ends will insert into an unreplicated target site indicated as the green/black triangle. Like standard *Ac/Ds* transposition, insertion of the *Ac/fAc* termini into the target site generates an 8-bp target site duplication (TSD; green/black triangle). (**C**) Insertion of the excised transposon termini places *fAc* and *fAc*-flanking DNA ahead of replication fork β (upper chromatid), and *Ac* and *Ac*-flanking DNA ahead of replication fork α to generate a rolling circle replicon (lower chromatid). DNA replication continues. (**D**) Following re-replication of *fAc*, *Ac*, and a portion of the flanking sequences, DNA replication forks α and β stall and abort, resulting in chromatids terminated by broken ends (the red > or < symbol) (**Michel et al., 1997**). The dotted red line connects the two broken ends that will fuse together. (**E**) Chromatid fusion produces a chromosome with two unequal sister chromatids: The upper chromatid contains a deletion of the segment from *fAc* to the *a/b* target site. The lower chromatid contains a TDD (left-hand loop), as well as a new CI (right-hand loop). The TDD contains the DNA deleted from the upper chromatid; the CI contains the re-replicated *Ac*, *fAc* and flanking sequences. The junction where broken chromatid ends were joined is indicated by the red ×.

The following figure supplement is available for figure 1:

**Figure supplement 1**. Reversed *Ac* ends transposition after DNA replication generates Tandem Direct Duplications (TDDs).

**Reversed *Ac* ends transposition generates tandem direct duplications and composite insertions**

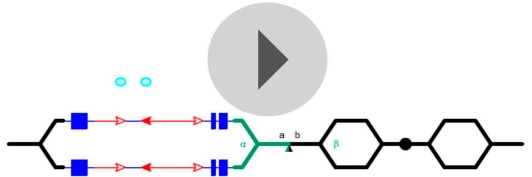

**Video 1**. Animation showing model for reversed Ac ends transposition during DNA replication. See ***Figure 1*** legend for details.

## Identification of alleles with Composite Insertions (CIs)

Both TDD and TDDCI alleles contain similar duplication structures and should exhibit similar phenotypes. Therefore, we screened maize ears as described previously to visually identify putative TDD-containing alleles (***Zhang et al., 2013***). We identified 25 candidate alleles, and cloned and sequenced the duplication/*Ac* junctions (the green segment flanking the *Ac* 5′ end in ***Figure 1E*** and ***Figure 2A***) from 16 of the 25 TDD/TDDCI candidates via *Ac* casting (***Singh et al., 2003***; ***Wang and Peterson, 2013***) or inverse PCR (iPCR) (See ***Zhang et al. (2013)*** for detailed screening and cloning methods). To identify the TDDCI alleles, we designed PCR primers that flank the progenitor insertion target sites for each allele (***Figure 2A***, primers 1 and 2). Primers 1 + Ac5 can amplify a product from both TDD and TDDCI while primers 2 + Ac3 can amplify a product only from TDDCI since the latter contains an additional CI (***Figure 2A***). As expected, PCR using primers 1 + Ac5 produced bands of the expected sizes in all the 16 alleles (***Figure 2B***, upper panel; seven examples are shown here). Whereas, primers 2 + Ac3 produced bands with expected sizes from only seven alleles (***Figure 2B***, lower panel). Sequencing of the PCR products obtained from primers 1 + Ac5 and 2 + Ac3 revealed that these seven TDDCI candidates have duplication/insertion breakpoints located from 13,392 bp to 1.7 Mb proximal to the *p1* locus on chromosome 1 (***Table 1***). Importantly, the *Ac* termini are flanked by 8-bp target site duplications (TSDs; green/black triangles in ***Figure 1E***) as predicted by the model in ***Figure 1*** (See ***Supplementary file 1*** for sequences containing TSDs).

Of particular importance are the results derived from three red/white twinned sectors, in which a sector of red kernel pericarp (seed coat) is twinned with an adjacent white pericarp sector (***Figure 3***). From each red pericarp sector, we isolated *P1-rr* alleles (*P1-rr-T21*, *P1-rr-T22* and *P1-rr-T24*), and from each white twin sector, we isolated corresponding *p1-ww* alleles (*p1-ww-T21*, *p1-ww-T22*, and *p1-ww-T24*). Similar types of twinned pericarp sectors have been shown to arise from the reciprocal products of standard *Ac* transposition events (***Greenblatt and Brink, 1962***; ***Chen et al., 1992***). Here, we propose that each pair of red/white twinned alleles are derived from the reciprocal TDDCI/deletion products of RET (sister chromatids shown in ***Figure 1E***). This was tested by PCR using primers 2 + Ac3; as shown in ***Figure 2B*** (lower panel), these primers produced bands of the same size for each set of twinned alleles. Moreover, for each pair of red/white co-twins, the sequences of the PCR products obtained using primers 2 + Ac3 are identical (***Supplementary file 1***). Together these results are consistent with the model of RET during DNA replication as shown in ***Figure 1***.

### Structures of the TDDCI alleles and Composite Insertions

Because PCR only provides information on rearrangement junctions, we further analyzed the structures of the candidate TDDCI alleles by DNA gel blot. Genomic DNA was digested with *SacI* and the blot was hybridized with probe 8B (gray boxes in ***Figure 2A***). This probe detects the *p1* gene (12.7 kb band), the paralogous *p2* gene (4.7 kb band), and the *p1-ww[4Co63]* allele (5.0 kb band) (***Goettel and Messing, 2010***) on the homologous chromosome. First, the 12.7 kb *p1* band is absent in the three twinned *p1-ww* alleles (*p1-ww-T22*, *p1-ww-T24*, and *p1-ww-T21*; ***Figure 2C***, lanes 4, 6 and 10, respectively). This result confirms the presence of a deletion as predicted by the model shown in ***Figure 1***. Second, the alleles *P1-rr-T24*, *P1-rr-E17*, and *P1-rr-E340* show a more intense 4.7 kb *p2* band in comparison with the 5.0 kb band (***Figure 2C***, lanes 5, 7, 8). This result is also expected because these three alleles have duplications of >70 kb (***Table 1***) that generate additional copies of the *p2* gene located ~70 kb proximal to *p1*. Third, alleles *P1-rr-T22*, *P1-rr-T21*, and *P1-rr-E5* (***Figure 2C***, lanes 3, 9 and 12, respectively) exhibit one or two new bands hybridizing with probe 8B. This is consistent with the presence of a CI that contains a newly-generated copy of the 8B sequence (***Figure 2A***). In *P1-rr-T22*, the duplication/insertion breakpoint occurred in the *p2* band containing probe 8B, resulting in a shift of the 4.7 kb band to ~8 kb (***Figure 2C***, lane 3). Moreover, this ~8 kb band is more intense than the 5.0 kb *p1-ww[4Co63]* band and the 12.7 kb *p1* band in *P1-rr-T22* (lane 3 in ***Figure 2C***). The model in ***Figure 1*** and our analyses indicate that the intense ~8 kb band is actually a triplet containing two copies of a

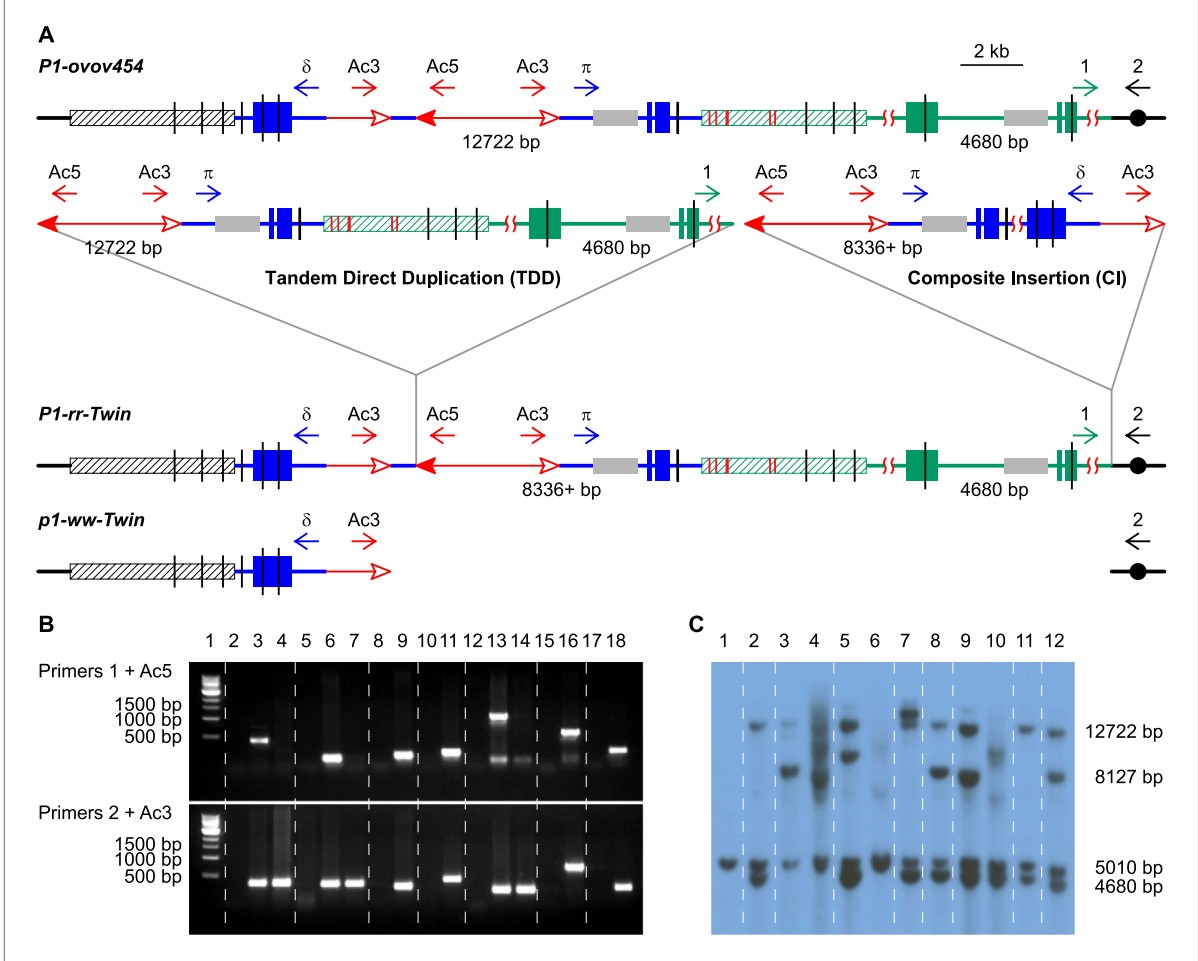

**Figure 2**. PCR screening and DNA gel blotting of candidate TDDCI alleles. (**A**) Detailed structures of *P1-ovov454* (progenitor) and RET-generated *P1-rr-twin/p1-ww-twin* (TDDCI/Deletion) alleles deduced from *Figure 1*. The horizontal blue lines are *p1* gene sequence while the green lines are *p1* proximal sequences, including the *p2* gene sequence (a *p1* paralog, ~70 kb proximal to *p1*); the blue and green boxes are exons 1, 2, and 3 (right to left) of *p1* and *p2*, respectively. The small horizontal arrows indicate the orientation and the approximate position of the PCR primers. The gray boxes indicate probe 8B used in DNA gel blot analysis, the short vertical black lines are *Sac*I sites, and the numbers between the *Sac*I sites indicate the lengths of those fragments detected by probe 8B. The hatched boxes represent the distal (black) and proximal (green) 5248 bp repeats flanking the *p1* locus. These repeats are identical except for six SNPs, indicated by short red vertical lines inside the green hatched box (SNPs 3 and 4 are only 43 bp apart). Other symbols have the same meaning as in *Figure 1*. (**B**) PCR products obtained using primers 1 + Ac5 (upper) or 2 + Ac3 (lower). Lane 1, 1 kb DNA ladder; lane 2, *P1-ovov454*; lane 3, *P1-rr-T22*; 4, *p1-ww-T22*; lane 5, *P1-ovov454*; lane 6, *P1-rr-T24*; 7, *p1-ww-T24*; lane 8, *P1-ovov454*; lane 9, *P1-rr-E17*; lane10, *P1-ovov454*; lane 11, *P1-rr-E340*; lane 12, *P1-ovov454*; lane 13, *P1-rr-T21*; 14, *p1-ww-T21*; lane 15, *P1-ovov454*; lane 16, *P1-rr-E5*; lane 17, *P1-ovov454*; lane 18, *P1-rr-E311*. Note: the sequences of primers 1 and 2 are specific for each allele. (**C**) DNA gel blot analysis of the TDDCI/deletion alleles. Genomic DNA was digested with *Sac*I and the blot was hybridized with probe 8B (see *Figure 2A* for the position of the probe). Lane 1: *p1-ww[4Co63]*, lane 2: *P1-ovov454/p1-ww[4Co63]*, lane 3: *P1-rr-T22/p1-ww[4Co63]*, lane 4: *p1-ww-T22/p1-ww[4Co63]*, lane 5: *P1-rr-T24/p1-ww[4Co63]*, lane 6: *p1-ww-T24/p1-ww[4Co63]*, lane 7: *P1-rr-E17/p1-ww[4Co63]*, lane 8: *P1-rr-E340/p1-ww[4Co63]*, lane 9: *P1-rr-T21/p1-ww[4Co63]*, lane 10: *p1-ww-T21/p1-ww[4Co63]*, lane 11: *P1-rr-E311/p1-ww[4Co63]*, lane 12: *P1-rr-E5/p1-ww[4Co63]*.

new 8461 bp *p1* fragment (one from the TDD, and a second from the CI, see below) and one copy of a 8127 bp *p2* fragment from the rearrangement junction. Further DNA gel blot analyses with a different *p1* probe (not shown) confirm that *P1-rr-T22* contains a TDD. All together, these results indicate that these four alleles—*P1-rr-T22*, *P1-rr-T24*, *P1-rr-E17*, and *P1-rr-E340*—contain the TDDCI structure.

We then characterized the structures of the CIs in the four TDDCI alleles. The model in *Figure 1* predicts that the insertion size and structure are determined by where re-replication aborts and how the resulting DSBs are repaired (*Figure 1D*). The structures of the CIs were determined by PCR using

**Table 1.** Features of alleles generated by RET-induced DNA re-replication

| Allele number | Allele type | Distance from donor locus to CI* |
|---|---|---|
| P1-rr-T21 | Solo-CI | 13,392 bp |
| P1-rr-E5 | Solo-CI | 16,497 bp |
| P1-rr-T22 | TDDCI | 70 kb |
| P1-rr-T24 | TDDCI | 80 kb |
| P1-rr-E340 | TDDCI | 447 kb |
| P1-rr-E311 | Solo-CI | 563 kb |
| P1-rr-E17 | TDDCI | 1.7 Mb |

*Distance given is from the 5′ end of Ac in the progenitor P1-ovov454 allele, to the point of insertion of the CI; that is, the distance between the TDD and CI insertion points in **Figure 2A**. In TDDCI alleles, this distance is also the length of the duplicated segment. Except for the fully sequenced alleles P1-rr-T21 and P1-rr-E5, the values given are based on the B73 reference genome sequence (**Schnable et al., 2009**), which likely differs from the genotype used in these experiments.

a series of divergent primer pairs flanking the Ac/fAc insertions (δ and π, the blue arrows in **Figure 2A**). These primers will not amplify products from the progenitor P1-ovov454 allele because they point away from each other (**Figure 2A**). However, if the CI is formed by re-replication and the Ac/fAc flanking segments are fused as shown in **Figure 1E** and **Figure 2A**, then these primers will be oriented towards each other and can amplify the internal sequence of the insertion. In this way, we obtained the internal sequences carried by the CIs in P1-rr-T22 and P1-rr-E17.

The CI in P1-rr-T22 is 23,238 bp in length (GenBank accession # KM013690), consisting of 14,484 bp of fAc and its distal flanking sequence and 8754 bp of Ac and its proximal flanking sequence (**Figure 4**); these two fragments are joined at a 4-bp microhomology sequence consistent with DSB repair via non-homologous end joining (NHEJ). In addition to the CI, the P1-rr-T22 allele carries a 70-kb TDD (**Table 1**), and its white co-twin p1-ww-T22 carries a reciprocal 70-kb deletion; moreover, the breakpoints of both the P1-rr-T22 duplication and p1-ww-T22 deletion contain 8-bp target site duplications. All of these features are predicted by the RET/re-replication model shown in **Figure 1**.

The CI in P1-rr-E17 (GenBank accession # KM013689) is 19,341 bp in length (**Figure 4**); its structure suggests that the DSBs predicted in **Figure 1D** were repaired via homologous recombination (HR) between two direct repeat sequences that flank the p1 gene in P1-ovov454 (**Lechelt et al., 1989**). These repeats (hatched boxes in **Figure 2A**) are 5248 bp in length; the proximal copy is 4555 bp from the Ac element while the distal copy is 2934 bp from fAc (**Figure 2A**). If re-replication continued beyond the Ac and fAc segments and into the flanking 5248 bp repeats before aborting, then the DSBs could be repaired via HR to generate the observed structures (**Figure 5**). The two repeat copies flanking P1-ovov454 differ at six SNPs in the distal half of the repeats (**Figure 2A**, red vertical short lines in the hatched box). Sequences of the P1-rr-E17 allele show that the repeat in the CI is identical to the proximal copy. These results suggest that the HR crossover occurred between the proximal halves of the two repeats (**Figure 5**).

For P1-rr-T24, no product could be amplified using the divergent primer strategy described above. However, a band of ~5.0 kb could be amplified using primers 1 + 2 which flank the insertion site. This band was sequenced and found to contain an intact Ac element (**Figure 4**). It seems very unlikely that this Ac was inserted through a simple transposition event, because the insertion site is located precisely at the duplication junction that is generated by RET, and an independent Ac transposition would not be expected to insert into precisely the same site. We suggest that the Ac insertion in P1-rr-T24 was produced by HR between the re-replicated Ac and fAc segments as they share 2039 bp of sequence identity (**Figure 6** and **Video 2**). Finally, the structure of the CI in P1-rr-E340 is still unknown; DNA gel blotting (not shown) indicated that the Ac-proximal fragment is in the range of 18–90 kb and the fAc-distal fragment is greater than 18 kb, resulting in a CI of at least 36 kb in length.

## RET-mediated DNA re-replication can generate solo-CI

PCR results show that the P1-rr-E311, P1-rr-T21, and P1-rr-E5 alleles contain junctions consistent with the presence of CI (**Figure 2B**, lanes 12–18). However, DNA gel blot analysis suggests that these same alleles do not contain TDDs (**Figure 2C**, lanes 9–12). Importantly, the CI in P1-rr-T21 is flanked by a target site duplication, and the CI insertion site is identical to the deletion breakpoint in the co-twin p1-ww-T21; these results strongly suggest that these twinned alleles were generated as the reciprocal products of an alternative transposition mechanism. We propose that the solo-CI alleles were formed by a mechanism similar to that shown in **Figure 1**, except that the termination of replication (**Figure 1C**)

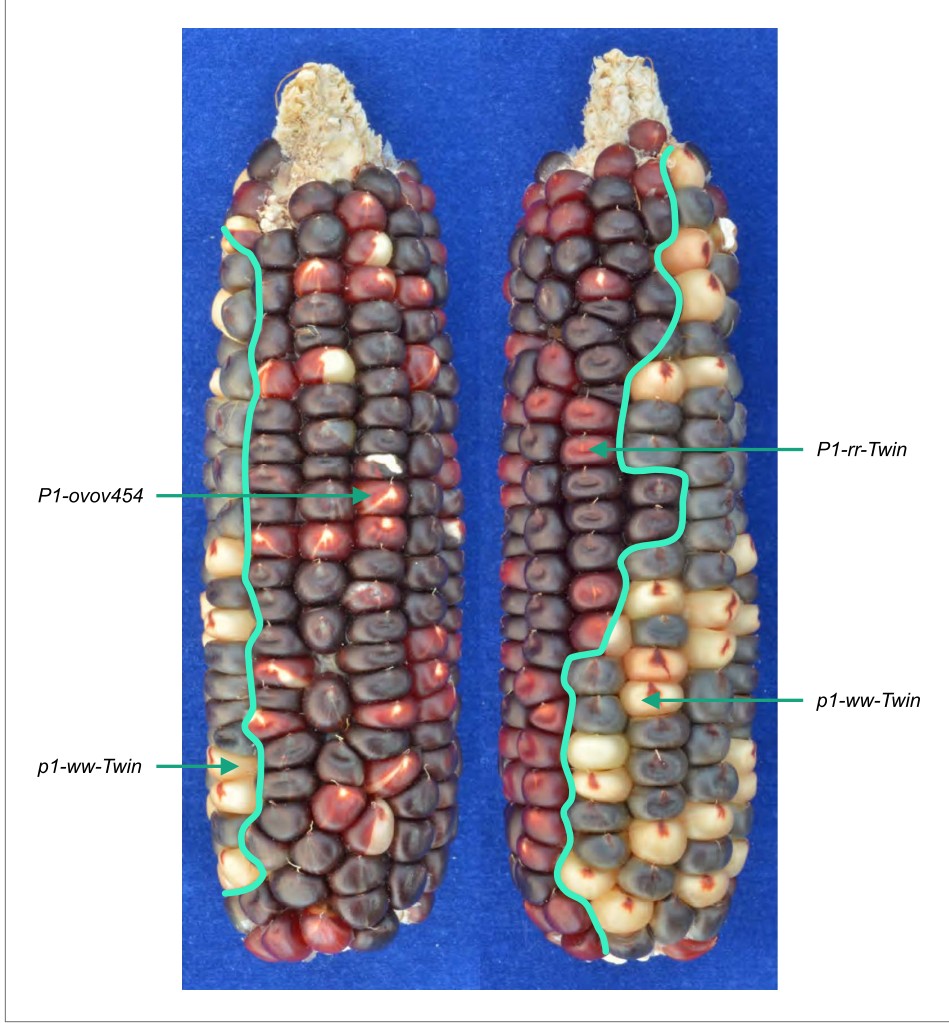

**Figure 3**. An ear with twinned sectors. The photo shows two sides of the same ear. Left-side view has a large area with parental *P1-ovov454* phenotype (orange pericarp with frequent colorless sectors), while the right-side view shows a large area with typical *P1-rr-Twin* phenotype (dark red pericarp with few colorless sectors). A single large *p1-ww-Twin* sector (kernels with mostly colorless pericarp) is visible in both views. The solid purple kernels present in all the sectors result from an independent germinal reversion of the *r1-m3::Ds* allele and can be ignored.

resulted in release and loss of the rolling circle. Because the TDD originates from the DNA included in the rolling circle, release of the rolling circle and subsequent DSB repair will result in a chromatid that carries only the CI (*Figure 7* and *Video 3*). The CI structures of these three alleles were characterized via PCR using primers δ and π as described above and are diagrammed in *Figure 4*.

In *P1-rr-T21*, the CI is 14,287 bp in length and contains a 3-bp microhomology region at the internal junction (GenBank accession # KM013688), consistent with DSB repair via NHEJ. For *P1-rr-E311*, the CI is 23,647 bp in length and has no apparent microhomology sequence at the internal junction (GenBank accession # KM013691), which is not uncommon for NHEJ-mediated repair (*Kramer et al., 1994*; *Wu et al., 1999*; *Lloyd et al., 2012*). *P1-rr-E311* does not contain a TDD, and its CI does not include fragment 8B; therefore the DNA gel blotting pattern in *P1-rr-E311* is the same as its progenitor *P1-ovov454* (lane 2 and lane 11 in *Figure 2C*). Finally, the CI in *P1-rr-E5* is 19,341 bp; its structure is identical to that in *P1-rr-E17* (*Figure 4*), even though these alleles arose independently and have the CI in different positions (16,497 bp and 1.7 Mb proximal to the *Ac* element in *P1-ovov454*, respectively; *Table 1*). We propose that both cases were produced via HR between the 5248 bp *p1*-flanking repeat sequences as described above and shown in *Figure 5*.

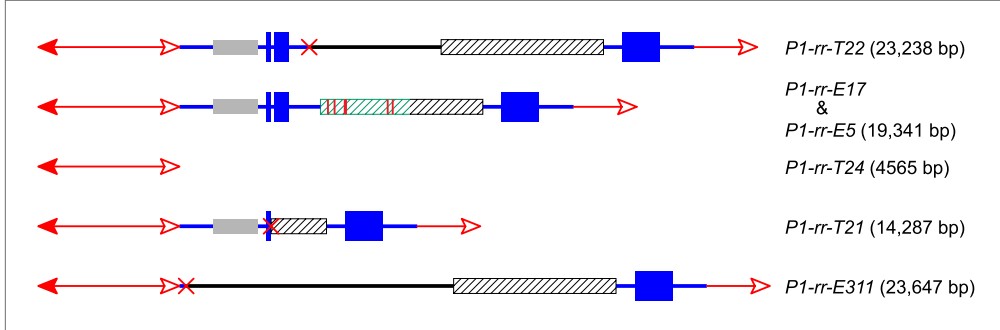

**Figure 4**. The structures and sizes of Composite Insertions (CIs). The double-headed arrows (left side) indicate *Ac* elements, while the single-headed arrows (right side) indicate *fAc*. The red × symbol indicates the junction of the two re-replicated segments in the insertion. Other symbols have the same meaning as in *Figure 1* and *Figure 2A*.

## RET mediates DNA re-replication at other genomic loci in maize

In addition to the above alleles, we identified another allele (*P1P2-3*, *Figure 8A*) that contains a CI but which was derived from a different progenitor allele (*p1-vv-D103*). The structure of *p1-vv-D103* is similar to that of *P1-ovov454*, except that the *fAc* element is shorter (779 bp vs 2039 bp in *P1-ovov454*) and the sequence distal to *fAc* has been replaced by chromosome 10 due to a chromosome 1–10 reciprocal translocation (in preparation). Like the examples described above, the *P1P2-3* allele arose in a single generation from *p1-vv-D103*; it contains a TDD of 80 kb, and a CI of 10,191 bp composed of 5017 bp of *Ac* and *Ac*-proximal flanking sequence and 5174 bp of *fAc* and *fAc*-distal flanking sequence. This structure is the same as that predicted by the model in *Figure 1*. The internal breakpoint junction of the CI contains a 9-bp homologous sequence, consistent with DSB repair via a microhomology-mediated end joining (MMEJ) mechanism (*Ma et al., 2003*; *McVey and Lee, 2008*).

If alternative *Ac/Ds* transposition can induce DNA re-replication and the formation of linked duplications and Composite Insertions, one may be able to detect these products at other loci. Interestingly, Barbara McClintock isolated an allele of the maize *bronze1* gene (*bz1-m4-D6856*) (*McClintock, 1956*) that has a complex structure consisting of three TDDs of *bz1* and its flanking sequence, separated by *Ds* elements (*Figure 8B*) (*Klein et al., 1988*; *Dowe et al., 1990*). The third repeat is not complete; its proximal side (including the *bz1* coding sequence) is truncated and joined to a truncated *Ds* sequence. This structure is similar to that of *P1-rr-T22*, *P1-rr-E17*, and *P1P2-3* described above: two intact Tandem Direct Duplications (*p1* vs *bz1* sequence), separated by TEs (*Ac* vs *Ds*), adjacent to a CI. In the case of *bz1-m4-D6856*, the CI contains the truncated copy of the tandem duplication and the truncated *Ds* and is flanked by 8 bp target site duplications. We propose that *bz1-m4-D6856* originated via a mechanism very similar to that shown in *Figure 1*: RET of two *Ds* elements located distal to the *bz1* gene, followed by insertion of the excised *Ds* termini into an unreplicated target site in the *bz1* 5′ UTR region. The three tandem repeats would have been formed by rolling circle replication; one replication fork would have dissociated from the circle distal to the *bz1* coding region to generate the incomplete repeat, while the other fork would have dissociated from the *Ds* element to generate a truncated *Ds* (*Video 4*). This model presupposes the existence of a *Ds* element (the leftmost element in *Figure 8B*) distal to the tandem repeats in *bz1-m4-D6856* and its progenitor allele. No such element was reported on the original *bz1-m4-D6856* genomic clones (*Klein et al., 1988*; *Dowe et al., 1990*). Efforts in our lab to identify a *Ds* element in this position in *bz1-m4-D6856* and related stocks have been unsuccessful. However, McClintock's description of the origin of *bz1-m4-D6856* (As reported in *Klein et al., 1988*) indicates that the *bz1-m4* progenitor produced a high frequency of dicentric chromosomes, while the *bz1-m4-D6856* derivative exhibited low dicentric frequency. Dicentric chromosome formation is a characteristic feature of alternative transposition reactions, such as RET, involving two nearby *Ac/Ds* elements (*Huang and Dooner, 2008*; *Yu et al., 2010*). The switch from high to low dicentric frequency observed by McClintock would be consistent with excision of the 'missing' *Ds* shortly after the formation of the *bz1-m4-D6856* allele.

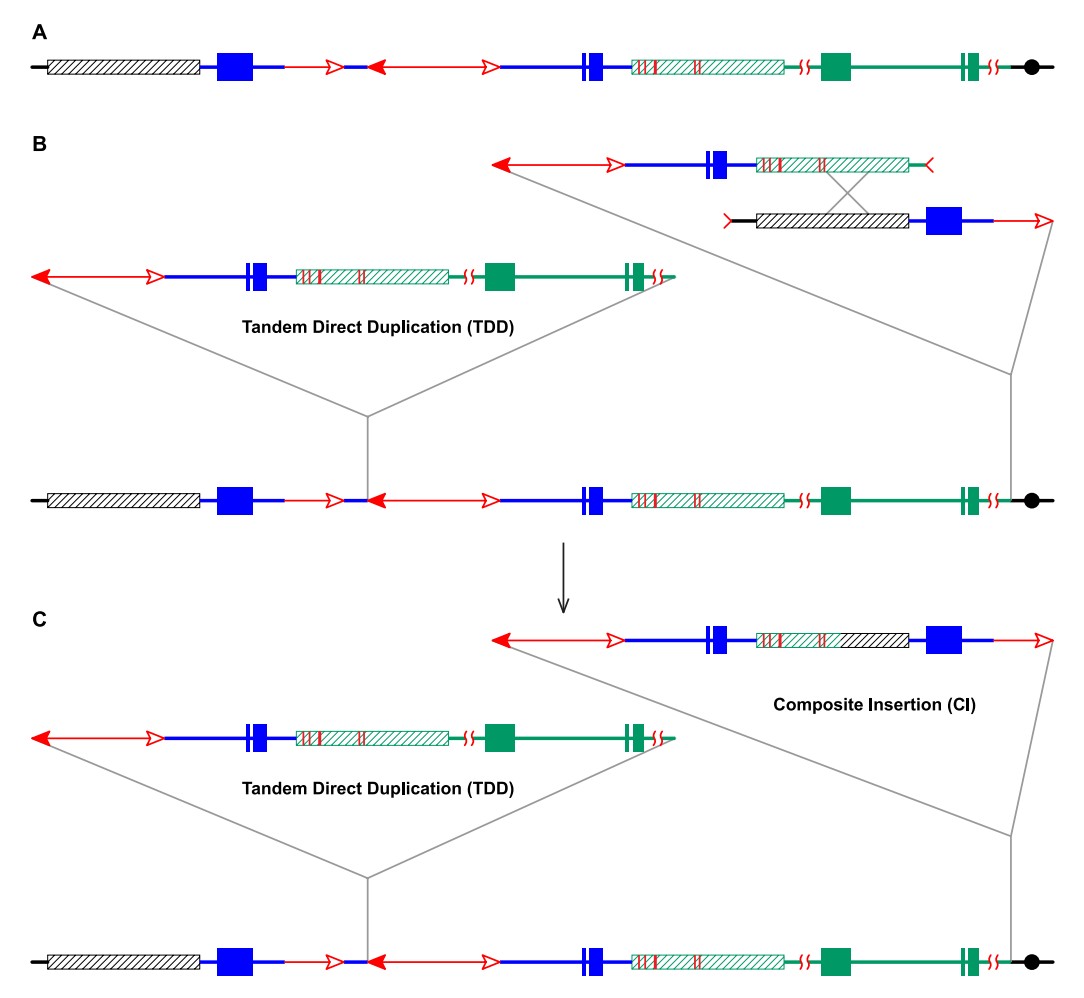

**Figure 5**. RET followed by homologous recombination generates identical 19,341 bp Composite Insertions in *P1-rr-E17* and *P1-rr-E5*. (**A**) Structure of the chromosome 1S segment containing the progenitor *P1-ovov454* allele, prior to RET. (**B**) Drawing shows the RET stage corresponding to *Figure 1D*. Recombination between the 5248 bp repeats near the two DSBs (marked by > or <) generates a Composite Insertion. (**C**) Structure of *P1-rr-E17* containing TDD (left-hand triangle) and Composite Insertion (right-hand triangle). All the symbols have the same meaning as in *Figure 2*. Note: *P1-rr-E5* contains the 19,341 bp CI but does not contain the TDD. See text for details.

## Discussion

We have identified a new pathway leading to re-replication of specific chromosome segments in maize. This pathway is initiated by transposase-induced excision of the replicated termini of nearby transposons, followed by insertion of the excised transposon ends into an unreplicated target site. Re-replication begins when chromosomal replication forks reach the transposon and may continue for considerable distances into the flanking DNA before aborting. The two resulting chromatid ends are joined together to restore chromosome linearity. This re-replication pathway is localized to the transposons and their flanking sequences and does not require origin re-initiation. In contrast, deregulating licensing factor activity results in re-firing of replication origin(s), leading to re-replication at multiple dispersed origins (*Green et al., 2006*).

Although little is known about termination of eukaryotic DNA replication, studies in yeast indicate that termination does not require specific terminator sites, but occurs wherever two replication forks converge (*McGuffee et al., 2013*). Here, we propose that alternative transposition reactions can interrupt normal fork convergence. For example, *Figure 1* shows that converging replication forks α and β are separated from each other by alternative transposition (*Figure 1C*); if not terminated by other factors, replication fork β could in principle continue until the end of the chromosome, which is ~48 Mb

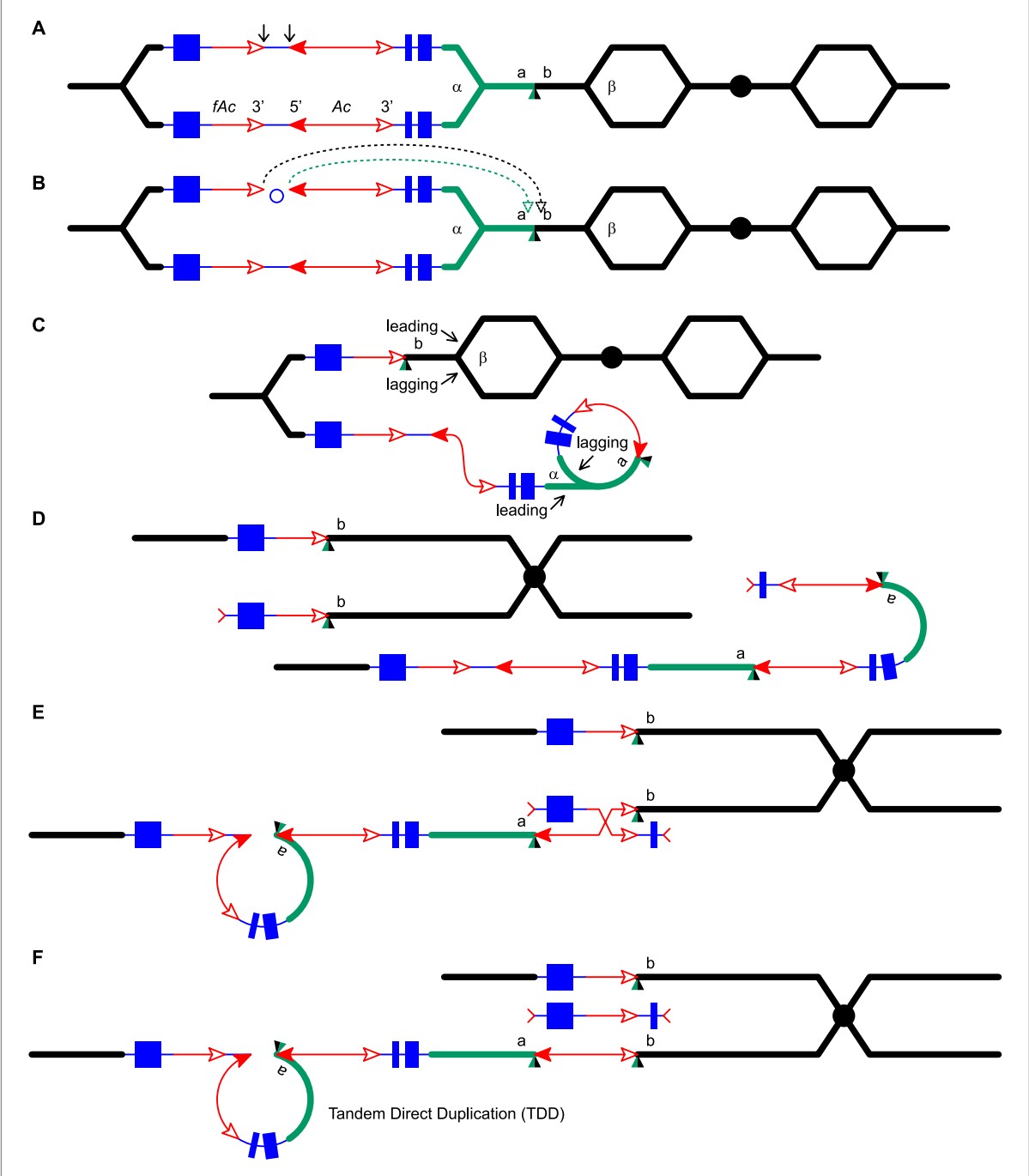

**Figure 6**. RET followed by homologous recombination generates a simple *Ac* insertion in *P1-rr-T24*. (**A**), (**B**), (**C**), and (**D**) are the same as in *Figure 1*. (**E**) Homologous recombination occurs between the re-replicated *Ac* and *fAc*. (**F**) Two new chromatids are formed: the lower chromatid contains a Tandem Direct Duplication and an *Ac* insertion, and the upper chromatid carries a reciprocal deletion. For animated version, see *Video 2*.

from the *p1* locus. However, our results suggest that DNA re-replication tends to abort after relatively short distances. The re-replicated segments generated from a single replication fork range in size from 4781 bp to 18,866 bp; the structure of the insertion in *P1-rr-E340* is unknown, but DNA gel blotting analysis suggests a size of at least 36 kb. Thus the total extent of DNA re-replication is less than 19 kb in eight of nine alleles examined. In contrast, break-induced replication in yeast is capable of replicating from the site of a DSB to the end of the chromosome (*Kraus et al., 2001*). What causes termination

**Reversed *Ac* ends transposition generates tandem duplications**

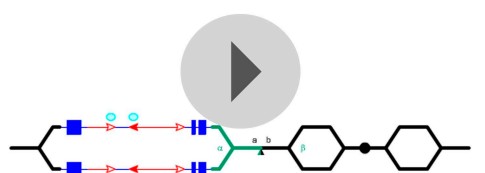

**Video 2**. Animation showing model for RET followed by homologous recombination and generation of a simple Ac insertion in P1-rr-T24. See text for details.

of re-replication following alternative transposition in maize? One possibility is fork chasing and head-to-tail fork collision (rear-ending), which has been shown to cause fork collapse and termination of DNA re-replication in Xenopus (*Davidson et al., 2006*). Alternatively, re-replication may spontaneously stall and abort due to compromised fork progression as reported in yeast (*Green et al., 2010*).

Our model proposes that DNA re-replication aborts to produce chromatids terminated by broken ends, which are joined together to restore chromosome linearity (*Figures 1, 6, 7, and 8*). If the chromatid DSBs were not repaired, the cell would die and that event would not be recovered in our screen. From a population of ~2000 plants, we isolated 16 alleles that carry a duplication and/or insertion structure. Nine of these 16 alleles (56%) have only a duplication (*Zhang et al., 2013*), which indicates that the target site was replicated at

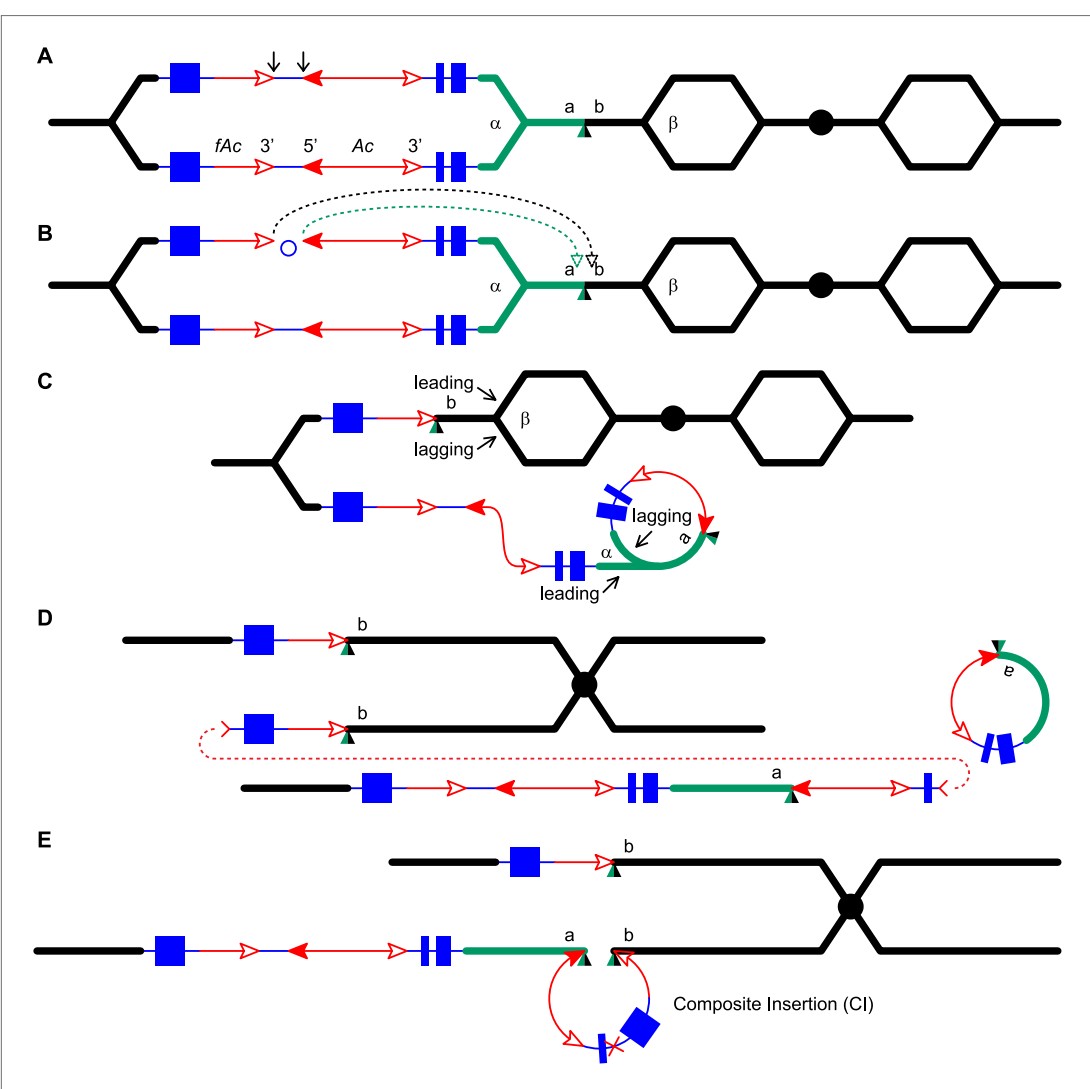

**Figure 7**. Generation of a Composite Insertion in the absence of a duplication. (**A**), (**B**), and (**C**) are the same as in *Figure 1*. (**D**) Upper chromatid contains deletion; in lower chromatid stalling and abortion of rolling circle replication fork releases the circle. (**E**) The two chromatids fuse to form a new chromatid containing a Composite Insertion. For animated version, see *Video 3*.

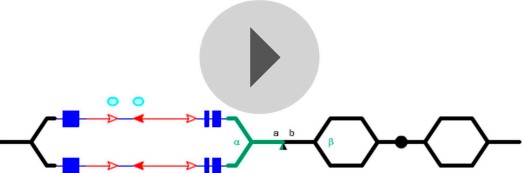

**Reversed *Ac* ends transposition generates solo composite insertions**

**Video 3**. Animation showing model for RET followed by NHEJ repair and generation of a CI in P1-rr-E311 and P1-rr-T21. The CI in *P1-rr-E5* was generated via a similar mechanism (i.e. the rolling circle was released when forming a DSB), but the DSBs were repaired by homologous recombination as shown in *Figure 5* (without the TDD). See text for details.

the time of RET (*Figure 1—figure supplement 1*); whereas seven alleles have an insertion, which indicates that the target site was unreplicated (*Figures 1, 6 and 7*). The frequency of insertion into an unreplicated target site is 7/16 (44%), which is similar to a previous estimate of *Ac* insertion into unreplicated sites (*Greenblatt and Brink, 1962*). Thus the products of insertion into unreplicated target sites are not significantly under-represented in our sample, suggesting that repair of re-replication-generated DSBs is quite efficient in mitotic S phase cells.

DNA lesions caused by replication fork stalling and collapse can be repaired by HR, NHEJ, MMEJ, replication slippage, FoSTeS (fork stalling and template switching), BIR (break-induced replication), MMBIR (microhomology-mediated

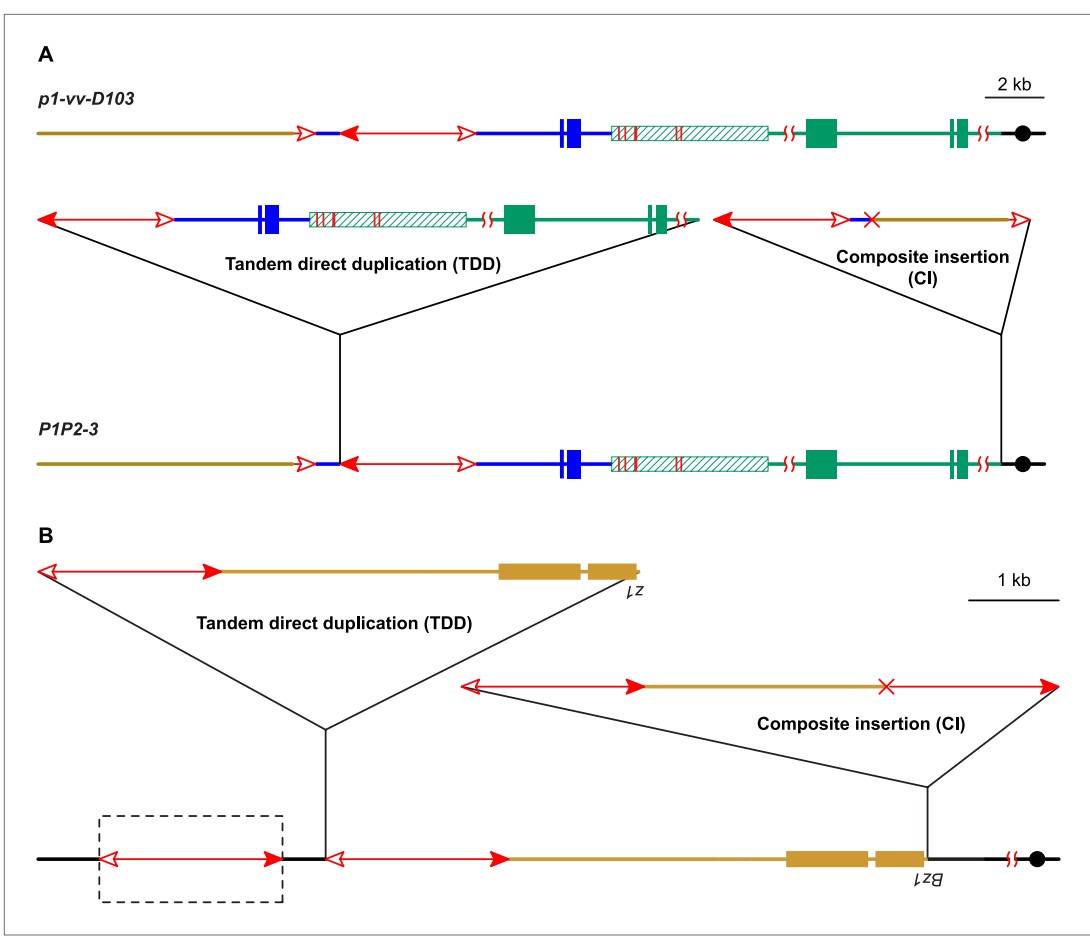

**Figure 8**. Two additional maize alleles likely generated by RET and re-replication. (**A**) Structure of progenitor allele *p1-vv-D103* (upper) and TDDCI allele *P1P2-3* (lower). The *p1-vv-D103* allele is carried on a chromosome 1–10 translocation; the brown line indicates DNA segment from chromosome 10. See text for details. Other symbols have the same meaning as in previous figures. (**B**) TDDCI structure of *bz1-m4-D6856*. The bronze-colored boxes indicate exons 1 and 2 (right to left) of the *bronze1* gene on maize chromosome 9. The baseline shows the predicted structure of the progenitor of *bz1-m4-6856*. The dashed box encloses a hypothetical *Ds* element proposed to have been involved in the generation of *bz1-m4-D6856* via RET. For animation, see *Video 4*. Other symbols as in previous figures. The structure of *bz1-m4-D6856* is deduced from *Dowe et al. (1990)* and *Klein et al. (1988)*.

**bz1-m4-D6856 was generated via reversed Ac ends transposition**

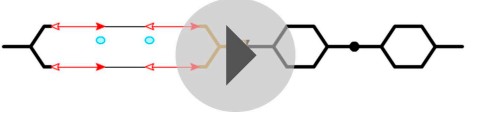

**Video 4**. Animation showing model for generation of TDDCI structure of bz1-m4-D6856 via rolling circle replication. See *Figure 8B* legend for details.

break-induced replication), MMIR (microhomology/ microsatellite-induced replication), and other mechanisms (*Kraus et al., 2001*; *Ma et al., 2003*; *Lee et al., 2007*; *McVey and Lee, 2008*; *Payen et al., 2008*; *Hastings et al., 2009a, 2009b*). In mammalian cells, replication fork-associated DSBs are predominantly repaired via HR (*Arnaudeau et al., 2001*). Among the six CI alleles sequenced here, three were repaired by HR and three by NHEJ, indicating that these two repair pathways have relatively similar activities during the S phase of mitosis in maize.

An important advantage of the maize system is the ability to identify genetically twinned sectors and to propagate and analyze their corresponding alleles. Because twinned alleles are the reciprocal products of a single event (*Greenblatt and Brink, 1962*), their structures should reflect a single parsimonious mechanism of origin. This allows us to distinguish among a variety of possible mechanisms for formation of segmental duplications. For example, non-allelic homologous recombination (NAHR) could generate a TDD joined and flanked by *Ac* as observed in *P1-rr-T24* if there were a *p1*-proximal *Ac* element in the progenitor allele *P1-ovov454* (*Figure 9*); however, such an NAHR event cannot explain the observed structure of the white co-twin *p1-ww-T24* (compare upper chromatids of *Figures 6F and 9C*). Similarly, re-replication-induced gene amplifications (RRIGA, a mechanism that couples NAHR and DNA re-replication) can also generate chromosome structures very similar to that of *P1-rr-T24* (*Green et al., 2010*; *Finn and Li, 2013*). Like NAHR, RRIGA would also require a *p1*-proximal *Ac* element as in *Figure 9B*. However, the reciprocal

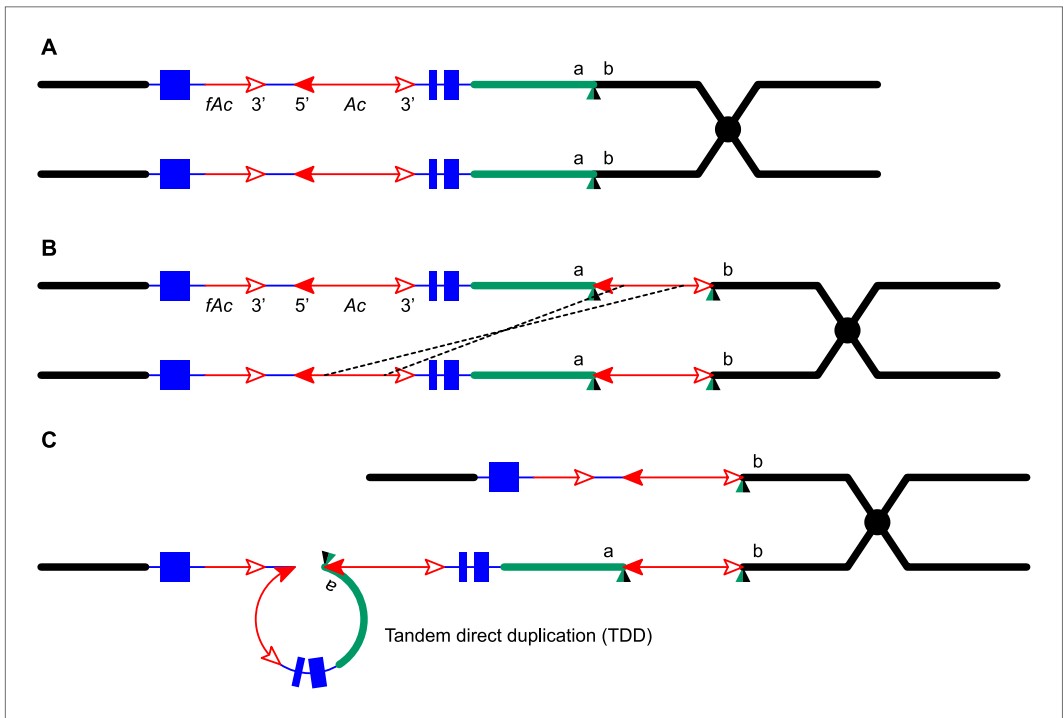

**Figure 9**. NAHR generates Tandem Direct Duplications. All the symbols have the same meanings as in *Figure 1*. (**A**) *Ac* transposes to a site between *a* and *b*. (**B**) Homologous recombination between two non-allelic *Ac* elements on sister chromatids generates a deletion (upper chromatid) and a TDD (lower chromatid) in (**C**).

product of an RRIGA-generated TDD would be a chromosomal fragment that lacks a centromere and telomeres, which would be lost in subsequent cell divisions. Therefore, neither NAHR nor RRIGA can generate the white co-twin *p1-ww-T24*. In contrast, the actual structure of the white co-twin *p1-ww-T24* is exactly as predicted by the RET re-replication model shown in *Figure 6*. The structures of the other TDDCI alleles are also inconsistent with NAHR and RRIGA: the duplicated segments are flanked by *Ac* on the left side and a CI on the right side (*Figure 1E*, lower chromatid), while NAHR/RRIGA-induced duplications would be flanked by identical *Ac* copies (*Figure 9C*, lower chromatid). Finally, NAHR and RRIGA generate TDDs of the same structure recurrently. In contrast, all of the TDDCI alleles we have isolated to date have different duplication breakpoints. This is consistent with their origin via alternative transposition, because the duplication endpoints are determined by the position of the transposon insertion site, which is expected to differ for each transposition event. Moreover, the RET reinsertion sites have the same characteristic features as for standard *Ac/Ds* transposition, including preferential insertion into nearby, hypomethylated, gene-rich regions (*Greenblatt and Brink, 1962*; *Chen et al., 1992*; *Vollbrecht et al., 2010*), and formation of 8-bp Target Site Duplications lacking sequence specificity (*Vollbrecht et al., 2010*). Taken together, our results consistently support the proposed mechanism of alternative transposition, re-replication, and repair.

In summary, we show here that reversed *Ac* ends transposition can generate TDDs and CIs. The TDDs range in size from several kb to >1 Mb and thus can increase the copy number of multiple linked genes and their regulatory sequences. The CIs we have discovered may be 20 kb or more in length. These are produced as a consequence of *Ac* transposition during DNA replication, and they exhibit a number of interesting features. First, the internal portions contain sequences that were originally flanking the donor *Ac/fAc* elements; the relative positions of these sequences are now switched, and they are fused together at a new junction. Because *Ac/Ds* elements are commonly inserted within or near genic sequences in plants, CI formation may shuffle the coding and/or regulatory sequences of the formerly flanking genes to create novel products. Moreover, the CIs are bordered by transposition-competent *Ac/fAc* 5′ and 3′ termini; hence, the entire CI has the structure of a macrotransposon (*Huang and Dooner, 2008*; *Yu et al., 2010*) that could subsequently transpose to new sites and increase in copy number. Eukaryotic genomes contain significant portions of Tandem Direct Duplications, dispersed segmental duplications, and tandem multi-copy arrays (*Bailey et al., 2003*, *2004*; *Rizzon et al., 2006*; *Shoja and Zhang, 2006*; *Bailey et al., 2008*; *Dujon, 2010*; *Tremblay Savard et al., 2011*); our results suggest that transposition-induced DNA re-replication may have played an important role in generating these segmental expansions during genome evolution.

## Materials and methods

Genetic stocks, other materials and methods used here are similar to those described previously (*Zhang et al., 2013*). Following is a condensed description, for full details see *Zhang et al. (2013)*.

### Genetic stocks

The maize *p1* gene encodes an R2R3-Myb transcription factor that regulates kernel pericarp (seed coat) and cob coloration. The phenotype conferred by each *p1* allele is indicated by the particular suffix: *P1-rr* specifies r̲ed pericarp and r̲ed cob, *p1-ww* specifies w̲hite (colorless) pericarp w̲hite (colorless) cob, and *P1-ovov* specifies o̲range v̲ariegated pericarp and o̲range v̲ariegated cob. *P1-ovov454* confers orange/red pericarp with frequent colorless sectors attributed to alternative transposition events that abolish *p1* function (*Yu et al., 2011*). The *p1-ww[4Co63]* allele is from the maize inbred line 4Co63 (*Goettel and Messing, 2010*). Ears of plants of genotype *P1-ovov454/ p1-ww[4Co63]* were fertilized with pollen from plants of genotype *C1, r1-m3::Ds, p1-ww[4Co63]*. The *r1-m3::Ds* allele is an *Ac* reporter allele: *Ac*-encoded transposase excises *Ds* from *r1-m3::Ds*, resulting in *r1* reversion and purple aleurone sectors. Changes in *Ac* copy number can be inferred by the negative *Ac* dosage effect: increased copy number of *Ac* delays the developmental timing of *Ac/Ds* transposition and reduces the frequency of early transposition events, generally producing variegated patterns with fewer, later transposition events (*McClintock, 1948*, *1951*). Reversed *Ac* ends transposition (*Figure 1*) can generate two non-identical sister chromatids: one carries a TDDCI, and the other a reciprocal deletion (*Figure 1E*). At mitosis these chromatids will segregate into adjacent daughter cells, forming an incipient twinned sector. The sector with the deletion

**Table 2.** Primer sequences

| Primer 1 | P1-rr-T22 | CTGTGGTCGTCCTGCTCCG |
|---|---|---|
| | P1-rr-E17 | AGATTTGACAGAACAGCCCGCAC |
| | P1-rr-T24 | GGTCACGCCCATAATAAAACAATAC |
| | P1-rr-E340 | AACCCGTCTCATCATCATCAGTGT |
| | P1-rr-T21 | GGTTTGTTTGTGCTGCCTCC |
| | P1-rr-E311 | TCGTTCTCTGGTTGGTCGTCGT |
| | P1-rr-E5 | ATTGGTCCCTCCCTCTCCCT |
| Primer 2 | P1-rr-T22 | AGAACTACTGGAACTCGCACCTCA |
| | P1-rr-E17 | CCAGAGTATAGGGTCATGGAGCC |
| | P1-rr-T24 | GCGTCCTCTATCCATTCACTTTCA |
| | P1-rr-E340 | TTTATGAGCCGCTGAATCGC |
| | P1-rr-T21 | CCGATGCTCTTTTCCTTCTCTTCC |
| | P1-rr-E311 | GCGATGCTATCAGTTAGACCAGGC |
| | P1-rr-E5 | CGCCGAACTTTCACTGCTCTGCTA |
| Ac3 | | GATTACCGTATTTATCCCGTTCGTTTTC |
| Ac5 | | CCCGTTTCCGTTCCGTTTTCGT |

chromosome has lost *Ac* and exons 1 and 2 of the *p1* gene; loss of *Ac* and *p1* functions will specify kernels with colorless pericarp and no purple aleurone sectors. The sector with the duplication chromosome retains a functional *P1-ovov454* gene and three copies of *Ac*; the predicted kernel phenotype will be orange/red pericarp with fewer colorless pericarp sectors, and fewer/smaller kernel aleurone sectors. Similar twinned sectors can also be formed via the mechanism in *Figure 6 or 7*. Mature ears were screened for multi-kernel twinned sectors with these characteristics; kernels from selected sectors were grown and analyzed. Alleles derived from twinned sectors or whole ears are indicated by a '*T*' or '*E*', respectively, prior to the allele number.

### Genomic DNA extractions, DNA gel blot hybridizations

Total genomic DNA was extracted using a modified cetyltrimethylammonium bromide (CTAB) extraction protocol (*Porebski et al., 1997*). Restriction enzyme digestions and agarose gel electrophoresis were performed according to manufacturers' protocols and *Sambrook et al. (1989)*. DNA gel blots and hybridizations were performed as described (*Sambrook et al., 1989*), except hybridization buffers contained 250 mM $NaHPO_4$, pH 7.2, 7% SDS, and wash buffers contained 20 mM NaHPO4, pH 7.2, 1% SDS.

### PCR amplifications

Sequences of oligonucleotide primers are shown in *Table 2*; note that primers 1 and 2 are specific to each allele, depending upon the flanking sequences. PCR was performed using HotMaster Taq polymerase from 5 PRIME (Hamburg, Germany). Reactions were heated at 94°C for 2 min, and then cycled 35 times at 94°C for 20 s, 60°C for 10 s, and 65°C for 1 min per 1 kb length of expected PCR product, then 65°C for 8 min. In some reactions 0.5–1 M betaine and 4–8% DMSO were added to improve yield. PCR products were separated on agarose gels, purified and sequenced directly by the DNA Synthesis and Sequencing Facility, Iowa State University, Ames, Iowa, United States. *Ac* casting and inverse PCR were used to isolate sequences flanking *Ac* insertions; these were performed as described previously (*Zhang et al., 2009*).

### Acknowledgements

This research is supported by NSF award 0923826 to TP and JZ. We thank co-PIs David F Weber (Illinois State University) and Dan Nettleton (Iowa State University) for their participation in this NSF-supported project. We thank Terry Olson for technical assistance, and Douglas Baker for field work.

## Additional information

### Funding

| Funder | Grant reference number | Author |
|---|---|---|
| Division of Molecular and Cellular Biosciences | Research Award 0923826 | Jianbo Zhang, Thomas Peterson |

The funder had no role in study design, data collection and interpretation, or the decision to submit the work for publication.

### Author contributions

JZ, TZ, Conception and design, Acquisition of data, Analysis and interpretation of data, Drafting or revising the article; DW, Acquisition of data, Analysis and interpretation of data, Drafting or revising the article; TP, Conception and design, Analysis and interpretation of data, Drafting or revising the article

## Additional files

### Supplementary file

• Supplementary file 1. Sequences flanking the Composite Insertion in the TDDCI and solo-CI alleles and sequences flanking *fAc* in the deletion alleles.

### Major dataset

The following previously published dataset was used:

| Author(s) | Year | Dataset title | Dataset ID and/or URL | Database, license, and accessibility information |
|---|---|---|---|---|
| Maizesequence | 2013 | Data from: B73 RefGen_v3 | http://www.ncbi.nlm.nih.gov/assembly/GCA_000005005.5/ | Publicly available at the NCBI Assembly database. |

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
