## [Decision Letter]

Thank you for sending your work entitled “Transposition-mediated DNA
re-replication in maize” for consideration at *eLife*. Your
article has been favorably evaluated by Ian Baldwin (Senior editor) and 3 reviewers.

Yubin Li and Amar Klar were responsible for two of the three peer reviews of your
submission and have agreed to reveal their identity.

The Senior editor and the reviewers discussed their comments before we reached this
decision, and we have assembled the following comments to help you prepare a revised
submission.

The manuscript describes a detailed analysis of an alternative transposition induced DNA
re-replication in the best-studied transposon system to date, namely the Ac/Ds system of
maize. A model is described of DNA re-replication mediated by the alternative
transposition of Ac/fAc, which in addition to other well-documented chromosomal
rearrangements caused by transposition, may have contributed to the dynamics of gene
creation and genome expansion. All reviewers agreed that the manuscript was dense but
well-written, and that the data supports the main conclusions of work and provides
valuable insights into TE behavior. The addition of a new pathway contributing to genome
dynamic by re-replication of particular chromosome regions through alternative
transposition represents a significant advance in our understanding of transposon
proliferation and genome evolution.

The reviewers make a number of points listed below, which if addressed would help to
clarify the presentation and broaden the impact of your story. We look forward to your
revision and reading how you have responded to these points in a cover letter.

1) The authors argue the generality of TDDCI in the maize genome by a previously
characterized bz1-m4-D6856 allele, which was isolated by Barbara McClintock. The actual
sequences of bz1-m4-D6856 allele and its progenitor line would provide more solid
evidence to support the authors' arguments.

2) The authors argue about the reinsertion frequency of the excised fragment and make
efforts to compare with previously reports statistically. About this issue, the reviewer
has unpublished data from an enormous population and supports the lower reinsertion
value as the authors present in their manuscript. It is a bit wasteful to make big deal
out of this.

3) Is there any commonality to the sequence context or genetic/epigenetic context of the
RET insertion sites recovered?

---

## [Author Response]

*1) The authors argue the generality of TDDCI in the maize genome by a previously
characterized bz1-m4-D6856 allele, which was isolated by Barbara McClintock. The
actual sequences of bz1-m4-D6856 allele and its progenitor line would provide more
solid evidence to support the authors' arguments*.

We did try to detect a *Ds* insertion distal to *bz1* in
stocks containing *bz1-m4-D6856* and its presumed progenitor. First, we
requested and received seeds of the putative *bz1-m4-D6856* progenitor
from Dr. Anita Klein, who originally published on this allele (24). Unfortunately the seeds were very old and
failed to germinate. Next we requested and received seeds of related stocks from Dr.
Hugo Dooner, and these we were able to propagate; however, the results of several PCR
experiments using *bz1-* and *Ds-*homologous primers were
negative. This negative result is not too surprising, considering the variation in
possible positions of *Ds*, as well as the uncertainty in identification
of the actual progenitor. Moreover, the published account of the origin of
*bz1-m4-D6856* suggests that the *Ds* element may have
excised during or shortly after the formation of the allele. The detailed description is
provided in the following excerpt from Klein, A.S., Clancy, M., Paje-Manalo, L., Furtek,
D.B., Hannah, L.C., and Nelson, O.E., Jr (1988) “The mutation
*bronze-mutable 4 derivative 6856* in maize is caused by the insertion
of a novel 6.7-kilobase pair transposon in the untranslated leader region of the
*bronze-1* gene”, Genetics *120*, 779-790:

*“The origin of bz-ml D6856 is complex (*Figure 1*). McClintock (1952) observed that
a transposable element at one locus would “spread” to adjacent loci. In
the maize line she was studying, Ds, in the presence of Ac (Activator), caused
chromosome breaks immediately distal to the shrunken (sh) locus. From this stock,
McClintock isolated new mutable alleles of the flanking genes, C-I (dominant
colorless) or Bz. The original bz-m4 allele was isolated in that study (B.
McClintock, personal communication). This bz-m4 line was stably recessive for the
shrunken (sh) trait. Later McClintock demonstrated that recombination between sh and
bz in this stock was substantially reduced, indicating that the unstable bz-ml allele
arose concommitantly with a deletion of chromosomal material in the interval between
these loci (McClintock 1965; Dooner 1981). In the presence of Ac, the original bz-m4
allele formed dicentric chromosomes at a high frequency. In a subsequent generation
this bz-m4 reverted to a Bz’ allele (B. McClintock, personal communication).
This was unstable, indicating that a Ds element was near or at the Bz’ allele.
Subsequently, again with Ac present, a gamete from a Bz’-m plant, culture
#6771, mutated to an unusual dark bronze, recessive allele which also had a
reduced frequency of dicentric formation. This allele was bz-ml
D6856*.*”*

Revised *eLife* text: “This model presupposes the existence of a
*Ds* element (the leftmost element in Figure 8) distal to the tandem repeats in *bz1-m4-D6856* and
its progenitor allele. No such element was reported on the original
*bz1-m4-D6856* genomic clones (24; 10).
Efforts in our lab to identify a *Ds* element in this position in
*bz1-m4-D6856* and related stocks have been unsuccessful. However,
McClintock's description of the origin of *bz1-m4-D6856* (as
reported in [24]) indicates that
the *bz1-m4* progenitor produced a high frequency of dicentric
chromosomes, while the *bz1-m4-D6856* derivative exhibited low dicentric
frequency. Dicentric chromosome formation is a characteristic feature of alternative
transposition reactions, such as RET, involving two nearby *Ac/Ds*
elements (22; 56). The switch from high to low
dicentric frequency observed by McClintock would be consistent with excision of the
“missing” *Ds* shortly after the formation of the
*bz1-m4-D6856* allele.”

*2) The authors argue about the reinsertion frequency of the excised fragment and
make efforts to compare with previously reports statistically. About this issue, the
reviewer has unpublished data from an enormous population and supports the lower
reinsertion value as the authors present in their manuscript. It is a bit wasteful to
make big deal out of this*.

We thank the reviewer for relating their unpublished data regarding reinsertion values.
Per the reviewer’s suggestion we have shortened this section, while keeping the
main point that the repair of broken chromatid ends is surprisingly efficient.

Revised *eLife* text: “Our model proposes that DNA re-replication
aborts to produce chromatids terminated by broken ends, which are joined together to
restore chromosome linearity (Figures 1, 6, 7 and 9). If the chromatid DSBs were not repaired, the cell would die and
that event would not be recovered in our screen. From a population of ∼2000
plants, we isolated 16 alleles that carry a duplication and/or insertion structure. Nine
of these 16 alleles (56%) have only a duplication (61), which indicates that the target site was replicated at
the time of RET (Figure 1—figure supplement 1); whereas seven alleles have an insertion, which indicates the target site
was unreplicated (Figures 1, 6 and 7).
The frequency of insertion into an unreplicated target site is 7/16 (44%), which is
similar to a previous estimate of *Ac* insertion into unreplicated sites
(18). Thus the
products of insertion into unreplicated target sites are not significantly
under-represented in our sample, suggesting that repair of re-replication-generated DSBs
is quite efficient in mitotic S phase cells.”

3) Is there any commonality to the sequence context or genetic/epigenetic
context of the RET insertion sites recovered?

In all respects, the RET insertion sites appear to be similar to those observed for
standard *Ac/Ds* transposition. We have included a sentence to this
effect in the Discussion:

“Moreover, the RET reinsertion sites have the same characteristic features as for
standard *Ac/Ds* transposition, including preferential insertion into
nearby, hypomethylated, gene-rich regions (18; 8;
53), and formation of
8-bp Target Site Duplications lacking sequence specificity (53).”